# High-Throughput Phenotyping Toolkit for Characterizing Cellular Models of Hypertrophic Cardiomyopathy In Vitro

**DOI:** 10.3390/mps2040083

**Published:** 2019-10-26

**Authors:** Diogo Mosqueira, Katarzyna Lis-Slimak, Chris Denning

**Affiliations:** Department of Stem Cell Biology, Centre of Biomolecular Sciences, University of Nottingham, Nottingham NG7 2RD, UK; mszkl@exmail.nottingham.ac.uk (K.L.-S.); plzcnd@exmail.nottingham.ac.uk (C.D.)

**Keywords:** disease modelling, hypertrophic cardiomyopathy, phenotyping, high-throughput, hypertrophy, high-content imaging, mitochondrial respiration, cardiomyocytes, cellular models, drug screening

## Abstract

Hypertrophic cardiomyopathy (HCM) is a prevalent and complex cardiovascular disease characterised by multifarious hallmarks, a heterogeneous set of clinical manifestations, and several molecular mechanisms. Various disease models have been developed to study this condition, but they often show contradictory results, due to technical constraints and/or model limitations. Therefore, new tools are needed to better investigate pathological features in an unbiased and technically refined approach, towards improving understanding of disease progression. Herein, we describe three simple protocols to phenotype cellular models of HCM in vitro, in a high-throughput manner where technical artefacts are minimized. These are aimed at investigating: (1) Hypertrophy, by measuring cell volume by flow cytometry; (2) HCM molecular features, through the analysis of a hypertrophic marker, multinucleation, and sarcomeric disarray by high-content imaging; and (3) mitochondrial respiration and content via the Seahorse™ platform. Collectively, these protocols comprise straightforward tools to evaluate molecular and functional parameters of HCM phenotypes in cardiomyocytes in vitro. These facilitate greater understanding of HCM and high-throughput drug screening approaches and are accessible to all researchers of cardiac disease modelling. Whilst HCM is used as an exemplar, the approaches described are applicable to other cellular models where the investigation of identical biological changes is paramount.

## 1. Introduction

Hypertrophic cardiomyopathy (HCM) is an intricate disease affecting 1:500 individuals where heart dysfunction often leads to sudden cardiac death [1]. The complexity of HCM encompasses a wide range of disease hallmarks as well as clinical manifestations and outcomes, hampering the development of efficient pharmacological treatment options [2]. Modeling HCM in vitro leads to a better understanding of disease progression towards unveiling novel molecular mechanisms and paving the way for targeted drug therapy [3]. 

Various models based on tissue and cellular explants, skinned myofibrils, and actin/myosin preparations have been generated, greatly contributing to uncover disease hallmarks and facilitating the development of new drugs, with some reaching clinical trials [4]. However, these disease models frequently produce contradictory results due not only to the recognized disease complexity (e.g., mutation-specific effects), but also owing to fundamental differences and limitations between models, such as low availability and demanding logistics of heart explants, over-simplicity of protein preparations, and species-differences relative to animal models [4]. 

Remarkably, cellular models enabled multi-parametric evaluation of numerous features of HCM, not only in cardiomyocytes derived from biopsies, but mostly in human pluripotent stem cell-derived cardiomyocytes (hPSC-CMs), which can be generated in unlimited numbers and recreate the patient’s genome [5]. These have paved the way for high-throughput drug screening approaches aimed at identifying new compounds with therapeutic potential [6]. Moreover, genome editing technologies such as CRISPR/Cas9 resulted in a more profound comprehension of genetic causation of HCM in hPSC-CMs, as mutations in sarcomeric genes often underlie disease progression in HCM patients [7,8]. This is a growing field, and recent isogenic cellular models enabled detailed description of disease progression in vitro, envisioning the development of new therapeutics [9,10,11,12,13,14]. Thus, there is a need for establishing a simple and straightforward toolkit for phenotyping HCM in vitro, to accelerate the characterization of disease hallmarks of different cellular models and expedite high-throughput drug screening approaches.

Herein, we describe in detail three protocols to phenotype 2D cellular models of HCM in vitro, focused on the analysis of hypertrophy, molecular features of disease, and bioenergetics. Altogether, these protocols consist of high-throughput and non-subjective tools to evaluate HCM disease hallmarks [9,14], indispensable for in-depth investigation of molecular mechanisms of disease underlying pharmacological intervention strategies. They are applicable to any (cardiac) cell-type but were optimized for the purpose of modelling HCM by using hPSC-CMs, available either commercially or via in-house differentiation protocols. 

The first protocol is aimed at evaluating hypertrophy (i.e., increase in cell size) using label-free flow cytometry of live cells. The overwhelming majority of hypertrophy analysis in vitro relies on the measurement of cell area, which is greatly affected by several parameters of 2D culture such as substrate properties [15], time in culture [16], and serum supplementation [17]. This method of flow volumetry relies on defining forward scatter (FSC) values of spherical beads of known diameters towards establishing a calibration curve. The FSC of freshly dissociated cardiomyocytes in suspension can then be analysed to determine their volume, in a high-throughput (100,000–250,000 cells/sample), unbiased, and statistically powerful flow cytometry method. This technique overcomes the pitfalls of 2D analysis and time-consuming imaging modalities of 3D stacking where the analysis of a limited number of cells requires acquisition of a high number of images [18]. Although this protocol demands immediate processing of freshly dissociated cardiomyocytes, it is inexpensive and only requires access to a standard flow cytometer. 

The second protocol consists of quantifying known molecular hallmarks of HCM by automated high-content imaging, enabling high-throughput evaluation of pathological markers by immunofluorescence. While several methods exist to undertake such analysis [19,20,21], they vary widely on sample preparation procedures [22], and often lack detailed methodological information needed to accurately acquire, and mostly quantify, fluorescent signal. Importantly, we have developed powerful algorithms to determine: (i) Multinucleation, (ii) expression of hypertrophic marker brain natriuretic peptide (BNP), and (iii) sarcomeric disarray. These require a high-content imaging system and computationally demanding processing software, which are commonplace in any modern research institute. We provide detailed step-by-step guidance to recreate our algorithms in order to accurately measure the above-mentioned hallmarks of HCM. This facilitates deep molecular characterisation of HCM phenotypes in cellular models by staining only three proteins and the cell nucleus. 

The final protocol is based on the functional evaluation of mitochondrial respiration in cardiomyocytes, known to be affected in HCM (termed the ‘energy depletion model’ [23]). Cardiomyocyte culture conditions (e.g., density, substrate, time in culture) required to accurately determine normalised oxygen consumption values in the Seahorse™ platform [24] were optimized. In addition, a method for measuring mitochondrial content by quantifying mitoDNA (ratiometric mitochondrial/nuclear DNA qPCR) in cardiomyocytes was adapted from other cell types [25]. These techniques require standard laboratory equipment such as a routine real-time qPCR system, a fluorescent cell imager, and a Seahorse™ analyzer (launched 13 years ago and used in over 5000 publications since then). With the extensive optimization we explore herein, they enable complementary assessment of compensatory responses in cardiomyocytes, associated with energetic imbalance in HCM.

The information gained by these tools can be harnessed to narrow down the number of conditions (i.e., genetic mutants and/or drugs) meriting further investigation, based on the analysis of cardiomyocyte contractility, calcium handling, and voltage transients. The established technologies to evaluate these functional parameters (e.g., patch clamp electrophysiology, multi-electrode arrays, optical imaging, and traction force microscopy) typically rely on low- to medium-throughput assays using single cells, 3D organoid, and/or engineered heart tissues [5,26]. However, recent developments have enabled measurement of contractility across all these configurations, based on quantification of pixel displacement in high-speed movies using a publicly available software [27]. Remarkably, it is now even possible to measure action potentials, cytosolic calcium flux, and contractility simultaneously [28]. These tools have achieved 44%–78% drug predictivity scores in hPSC-CMs [29]. Moreover, despite the existence of challenges such as incomplete hPSC-CM maturity and lack of multicellular complexity, these approaches have been used to investigate the effects of genetic mutations in HCM progression, towards clarifying genotype-phenotype relationships [8].

Altogether, our three protocols are aimed at providing a fundamental characterisation of the main HCM phenotypes in cellular models (Figure 8, expected results), creating a platform for more in-depth studies into the molecular mechanisms governing disease progression, towards its treatment. 

## 2. Experimental Design

Accurate phenotyping of cellular models of HCM via this toolkit typically requires dissociation of 2D monolayers or 3D constructs into single-cell suspensions. Efficient isolation of cardiomyocytes from heart tissue biopsies has been the subject of numerous reports and typically consists of enzymatic bulk digestion [30], Langendorff method [31] or mechanical disruption procedures [32]. A recent protocol optimized this process in five main steps: 1) Myocardium dissection into 200 μm-thick slices; 2) slice perfusion with a Ca^2+^-free solution; 3) enzymatic digestion using collagenase II and protease XXIV; 4) filtration of cardiac tissue extract with a 100 μm mesh (to break cell clumps and minimize cell sampling biases [33]); and 5) in vitro culture in 5% foetal bovine serum-containing medium [34]. This method resulted in up to 65% viable cardiomyocyte isolation yield and enabled phenotypic studies of electrophysiology, Ca^2+^ imaging, and Seahorse™ analysis. Alternatively, cardiomyocytes can be efficiently sourced in high numbers through cardiac differentiation of hPSCs [35] followed by their dissociation into single cells using a collagenase II digestion protocol [36].

While these cell sources and their culture methods vary between laboratories, baseline conditions such as serum supplementation and time in culture should be kept constant between the different groups being compared (e.g., cell lines or treatments), to minimize technical artefacts. 

All cell culture manipulation is performed in a routine type II Biological Safety Cabinet, and cells are cultured in a humidified incubator, at 37 °C and 5% CO_2_. Once a single cell suspension is obtained, each of these 3 protocols can be performed (Figure 1).

While the hypertrophy analysis consists of direct assessment of freshly dissociated cardiomyocytes, the other 2 protocols require further cell culture upon replating. All protocols have built-in quality control steps (e.g., exclusion of cell debris and non-cardiomyocytes) for the cells being used, as well as normalization methods to account for differences in cell numbers. 

All the exemplary data reported in this manuscript (Figure 8) are based on the comparison between a hPSC-CM cell line bearing the R453C-β-myosin heavy-chain mutation (HCM line) and its isogenic wild-type control (healthy). hPSC lines were derived in-house [37] from material harvested under ethical approval from Nottingham Research Ethics Committee 2 (09/H0408/74) with informed patient consent.

### 2.1. Materials

#### 2.1.1. Hypertrophic Analysis

Polystyrene Particle Size Standard Kit, Flow Cytometry Grade (Spherotech, Lake Forest, IL, USA; Cat. no.: PPS-6K).Phosphate Buffer Saline (PBS, Gibco™, Fisher Scientific, Loughborough, UK; Cat. No: 14190-094).

#### 2.1.2. High-Content Imaging

CellCarrier-96 well plates (PerkinElmer, Waltham, MA, USA; cat. No.: 6005550)Phosphate Buffer Saline (PBS, Gibco™, Fisher Scientific, Loughborough, UK; Cat. No: 14190-094).Vitronectin (VTN-N) Recombinant Human Protein, Truncated (500 μg/mL in PBS) (Gibco™, Fisher Scientific, Loughborough, UK; Cat. No: A14700).RPMI 1640 medium (Gibco™, Fisher Scientific, Loughborough, UK; Cat. No: 21875034).B-27™ Supplement (50X), custom (Gibco™, Fisher Scientific, Loughborough, UK; Cat. No: 0080085SA).Endothelin-1 (10 μM in cell culture grade water) (ET1, Sigma-Aldrich, St. Louis, MO, USA; Cat. No.: E7764). Store at −80 °C for 1 year.Bosentan (100 μM in DMSO) (BOS, Sigma-Aldrich, St. Louis, MO, USA; Cat. No.: SML1265). Store at −80 °C for 1 year.Brefeldin A (1 mg/mL in methanol) (BFA, Sigma-Aldrich, St. Louis, MO, USA; Cat. No.: B7651). Store at −20 °C for 1 year.4% Parafolmadehyde (PFA, VWR International, Leicester, UK; Cat. No.: J61899.AP).Parafilm (VWR International, Leicester, UK; Cat. No.: HS234526A).Triton™-X-100 (0.1% *v/v* in PBS) (Sigma-Aldrich, St. Louis, MO, USA; Cat. No.: T8787). Store at room temperature for 1 year.Tween™-20 (0.1% *v/v* in PBS) (Fisher Scientific, Loughborough, UK; Cat. No: 10113103)Store at room temperature for 1 year.Goat Serum (4% *v/v* in PBS) (Sigma-Aldrich, St. Louis, MO, USA; Cat. No.: G9023).Anti α-actinin antibody raised in mouse (Sigma-Aldrich, St. Louis, MO, USA; Cat. No.: A7811).Anti-cardiac troponin-T antibody raised in rabbit (Abcam, Cambridge, UK; Cat. No.: 45932).Anti-proBNP antibody raised in mouse (Abcam, Cambridge, UK; Cat. No.: 13115).Alexa Flour-488 secondary antibody Goat anti-mouse (Thermo Fisher Scientific, Waltham, MA, USA; cat. No.: A11001).Alexa Flour-647 secondary antibody Goat anti-rabbit (Thermo Fisher Scientific, Waltham, MA, USA; cat. No.: A21244).HCS CellMask Deep Red Stain (Invitrogen, Carlsbad, CA, USA, Cat. No.: H32721).4′,6-diamidino-2-phenylindole (DAPI, Sigma-Aldrich, St. Louis, MO, USA; Cat. No.: D9542).

#### 2.1.3. Mitochondrial Respiration and Content 

Seahorse XF Cell Mito Stress Starter Pack 96-well format (Agilent, Santa Clara, CA, USA, Cat No.: 103708-100).VWR Signature™ 200 µL Pipet Tips, Graduated (VWR International, Leicester, UK; Cat. No.: 37001-532).bisBenzimide H 33,342 trihydrochloride (Hoechst 33342, Sigma-Aldrich, St. Louis, MO, USA; Cat. No.: 14533).1.7 mL microtubes (Axygen, Corning, Amsterdam, The Netherlands; Ctat. No.: MCT-175C).DNeasy Blood and Tissue Kit (Qiagen, Hilden, Germany; Cat. No.: 69504).Molecular-grade water (Sigma-Aldrich, St. Louis, MO, USA; Cat. No.: W4502).TaqMan™ Gene Expression Master Mix (Fisher Scientific, Loughborough, UK; Cat. No.: 4369016).MicroAmp™ Fast Optical 96-Well Reaction Plate, 0.1 mL (Applied Biosystems, Foster City, CA, USA, Cat. No.: 4346907).MicroAmp™ Optical Adhesive Film (Applied Biosystems, Foster City, CA, USA, Cat. No.: 4311971).TaqMan™ Gene Expression Assay (FAM) - MT-ND1 probe (Applied Biosystems, Foster City, CA, USA, Cat. No.: 4331182, Assay ID: Hs02596873_s1).TaqMan™ Gene Expression Assay (FAM) - MT-ND2 probe (Applied Biosystems, Foster City, CA, USA, Cat. No.: 4331182, Assay ID: Hs02596874_g1).TaqMan™ Gene Expression Assay (FAM) - ACTB probe (Applied Biosystems, Foster City, CA, USA, Cat. No.: 4331182, Assay ID: Hs03023880_g1).

### 2.2. Equipment

#### 2.2.1. Hypertrophic Analysis

FC500 Flow Cytometer (Beckman Coulter, Indianapolis, IN, USA), or equivalent.

#### 2.2.2. High-Content Imaging

Operetta™ High-Content Imaging System (PerkinElmer, Waltham, MA, USA; Cat. No.: HH12000000), or equivalent.Nikon’s Eclipse TE2000 Inverted Research Microscope (Nikon, Minato, Tokyo, Japan), or equivalent.

#### 2.2.3. Mitochondrial Respiration and Content

Non-CO_2_ oven set at 37 °C (Stuart Scientific, Staffordshire, UK; Cat. No.: S120H).Seahorse XFe96 Analyzer (Agilent, Santa Clara, CA, USA, Cat No.: S7800B).CellaVista™ Automated plate imager (Synentec, Elmshorn, Germany), or equivalent.7500 Fast Real-Time PCR System (Applied Biosystems, Foster City, CA, USA, Cat. No.: 4351106), or equivalent.Nikon’s Eclipse TE2000 Inverted Research Microscope (Nikon, Minato, Tokyo, Japan), or equivalent.NanoDrop™ Microvolume Spectrophotometer (ND-2000, Fisher Scientific, Loughborough, UK).

### 2.3. Software

#### 2.3.1. Hypertrophic Analysis 

Kaluza analysis v2.1 (Beckman Coulter, Indianapolis, IN, USA), or equivalent.Microsoft Excel™ (Microsoft, Redmond, WA, USA).GraphPad Prism v8.2.0 (La Jolla, CA, USA).

#### 2.3.2. High-Content Imaging

Harmony High-Content Imaging and Analysis Software (PerkinElmer, Waltham, MA, USA; cat. No.: HH17000001), with PhenoLOGIC™ machine-learning technology (required only for sarcomeric disarray analysis).GraphPad Prism v8.2.0 (La Jolla, CA, USA).

#### 2.3.3. Mitochondrial Respiration and Content

Seahorse Wave Desktop Software (Agilent, Santa Clara, CA, USA, https://www.agilent.com/en/products/cell-analysis/cell-analysis-software/data-analysis/wave-desktop-2-6).7500 Real-Time PCR Software (v2.3, Applied Biosystems, Foster City, CA, USA, https://www.thermofisher.com/uk/en/home/technical-resources/software-downloads/applied-biosystems-7500-real-time-pcr-system.html), or equivalent.Microsoft Excel™ (Microsoft, Redmond, WA, USA).GraphPad Prism v8.2.0 (La Jolla, CA, USA).

## 3. Procedure

### 3.1. Hypertrophy Analysis. Time for Completion: 1 Day

#### 3.1.1. Data Acquisition. Time for Completion: 2 h

Resuspend freshly dissociated cardiomyocytes in 500 μL Phosphate Buffer Saline (PBS) at 1 × 10^5^–2.5 × 10^5^ cells per sample and place them on ice prior to flow cytometry analysis.
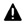
**CRITICAL STEP** Prior culture of cardiomyocytes in serum-containing medium may mask hypertrophic phenotypes [17], so exposure to serum should be minimized or fully eliminated.Add 2 drops of bead solution of each size to a flow cytometer tube, mix well by flicking it, and measure FSC and side scatter (SSC) values (area, width, and height) at 488 nm laser wavelength (area and height).
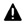
**CRITICAL STEP** Use of 100 μM flow cytometer nozzle size is highly recommended to minimize sampling biases [33].Run cell samples in the flow cytometer, using the same operating settings as those used for the calibration beads.
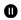
**PAUSE STEP** Once data from flow cytometry is acquired, it can be analysed at any point.

#### 3.1.2. Data Analysis. Time for Completion: 3 h


4.In the Kaluza software (or any equivalent flow cytometry analysis software), gate to remove debris and duplets/triplets (exclude events with very low FSC and SSC Area values) (Figure 2A,B).5.Gate to define FSC-A for each bead used for the calibration. Export FSC-A values relative to each gate (corresponding to each bead) to an excel file. Ensure there is consistency between the units being used for beads and samples (i.e., linear or log) (Figure 2C,D).6.In the Excel software, plot the FSC-A median values (y axis, obtained from each gate in the Kaluza software) relative to the known bead diameters (x axis). Add trendline of linear regression and display equation (FSCA = slope×diameter+intercept) and R-square value determining the calibration curve (Figure 2E).
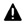
**CRITICAL STEP** If the R-squared value (measure of how much the variation of the FSCA is explained by variation of the bead size) is below 0.95, then the bead solutions must be thoroughly mixed to disrupt clumps.7.Gate samples to remove debris and duplets/triples. In cell samples, SSC-width is used in the x axis instead of FSC-A due to higher heterogeneity of the cardiomyocytes relative to the beads (Figure 2F–H).8.Export data from each sample from Kaluza to Excel, as done in 5.9.Using the calibration equation determined in 6, convert the FSC-A values measured for each sample into cell diameter: $diameter\;=\;\frac{\left(FSCA\;-intercept\right)\;}{slope}$.10.Determine the cell volume by applying the formula of the volume of a sphere: $V\;=\;\frac{\Pi }{6}\times d^{3}$, where d is the cell diameter determined in 9. This assumes the approximation of each cell in suspension to a sphere, which is also applicable to the calibration beads.11.Import the cell volume values into GraphPad software, as a column table (whereby each column represents a separate sample).12.Choose “violin plot only” in the “Box and plot” options to graphically represent the data distribution and the main statistical parameters (median and quartiles). Adjust colour as desired (Figure 8A).13.Perform statistical analysis (depending on the experimental design) to compare volume distribution between samples.**OPTIONAL STEP** Plotting such large sample sizes in GraphPad can be computationally demanding. Alternatively, a boxplot can be generated by inputting only 7 cardiomyocyte volume values per condition: Minimum, maximum; 25th percentile, 75th percentile, and the median three times. These values can be easily determined using Excel™ software (Figure 8B).


### 3.2. High Content Imaging. Time for Completion: 7 Days

#### 3.2.1. Replating Cardiomyocytes. Time for Completion: 2 h

Prepare a Vitronectin solution at 5 μg/mL in PBS (1:100 dilution from stock).Coat a CellCarrier-96 well plate with 50 μL/well of Vitronectin at 5 μg/mL. Incubate for 1 h at room temperature (RT).Supplement RPMI 1640 medium by adding B27 supplement (RB27, 50x dilution), aliquot volume needed and let it reach RT.**OPTIONAL STEP** Other extracellular matrix proteins and culture media could be used instead of Vitronectin and RB27, provided they were previously tested for the specific cardiomyocyte source in culture.Aspirate Vitronectin and add 50 μL of PBS per well. Aspirate PBS and add 75 μL of RB27 per well.Plate cardiomyocytes at approximately 110,000 cells/cm^2^ (35,000 cells per well).**OPTIONAL STEP** Allow the cells to settle for 0.5 h at room temperature before placing them in the incubator, to minimize edge effects due to the formation of convection currents because of temperature differences between the medium and the incubator.

#### 3.2.2. Culturing and Fixing Cardiomyocytes for Immunostaining. Time for Completion: 4 Days

6.Replace cardiomyocyte medium (100 μL/well) every 2 days for 4 days.7.(a) For multinucleation and sarcomeric disarray quantification, fix cells by incubating with 50 μL/well 4% PFA for 15 min at RT, after washing cardiomyocytes with PBS. Wash again after fixation and add 200 μL PBS/well.(b) To determine expression of hypertrophic marker BNP, replace medium with either 100 μL of RB27 (untreated control), RB27 supplemented with 10 nM Endothelin-1 (ET1, 1,000X dilution), or RB27 supplemented with 100 nM Bosentan (BOS, 1000X dilution) and incubate for 15 h at 37 °C, 5% CO_2_. Prepare a 5 μg/mL BFA solution in RB27 (200X dilution from the stock) and add 25 μL/well to the remaining 100 μL of RB27 (+/- ET1 or BOS). Incubate cells for 3 h at 37 °C, 5% CO_2_ followed by fixation in PFA as described in 7a.
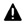
**CRITICAL STEP** As controls for immunostaining, some wells should be included for staining with the secondary antibody only to detect unspecific binding (secondary-only control). Alternatively, a cell-type control based on culturing non-cardiomyocytes (e.g., fibroblasts) in the same plate type can also be used and immunostained the same way as cardiomyocytes (Figure A1).
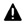
**CRITICAL STEP** BFA treatment prevents BNP secretion in cardiomyocytes, reaching maximal levels at 15 h post ET1 treatment. ET1 and BOS treatments are required to threshold BNP signal intensity (Figure A2).
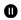
**PAUSE STEP** Fixed cell plates can be stored at 4 °C for up to 2 months prior to immunostaining and imaging. Wrap plates in parafilm to prevent PBS evaporation and consequent plate drying.

#### 3.2.3. Immunostaining Cardiomyocytes. Time for Completion: 2 Days

Reagent volume per well: 50 μL.

8.Wash fixed cells with PBS. Permeabilize cells by treating with 0.1% *v/v* Triton-X and incubating at RT for 15 min.9.Wash 3 times with PBS.10.Add 4% *v/v* goat serum in PBS (blocking solution). Incubate for 1 h at RT.11.Replace blocking solution by primary antibodies pertaining each staining:
Multinucleation and sarcomeric disarray: Anti α-actinin antibody (1:800 in blocking solution).Hypertrophic marker: Anti cardiac troponin-T and anti proBNP4 (both at 1:500 in blocking solution).

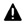
**CRITICAL STEP** The secondary-only controls should be incubated with blocking solution instead. Cell-type controls should be treated the same way as cardiomyocytes.12.Incubate for 15 h at 4 °C.13.Wash 3 times in 0.1% *v/v* Tween-20.14.Replace washing solution with secondary antibodies pertaining each staining. Incubate for 1 h at RT:
Multinucleation and sarcomeric disarray: Alexa Fluor-488 goat anti-mouse (1:400 in blocking solution).Hypertrophic marker: Alexa Fluor-488 goat anti-mouse and Alexa Fluor-647 anti-rabbit (both at 1:400 in blocking solution).
15.Wash 3 times with 0.1% *v/v* Tween-20 and incubate for 5 min at RT.16.Add nuclei/cytoplasm counterstains. Incubate for 30 min at RT:
Multinucleation and sarcomeric disarray: HCS Cell Mask Deep Red (1:10,000 in PBS) and DAPI (1:500 in PBS).Hypertrophic marker: DAPI (1:500 in PBS).
17.Wash 2x with PBS. Add 200 μL PBS per well prior to analysis or storage.
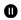
**PAUSE STEP** Immunostained cell plates can be stored at 4 °C for up to 2 weeks prior to imaging. Wrap plates in aluminium to prevent PBS evaporation and consequent plate drying, and to protect from light.

#### 3.2.4. Imaging Immunostained Cardiomyocytes. Time for Completion: 2 h per Plate

18.Initiate Operetta™, light source, and Harmony software for image acquisition.19.Select “96 PerkinElmer CellCarrier” as plate type, “20x long WD” as Objective, “Non-confocal” as Optical Mode, and 100% excitation intensity.**OPTIONAL STEP** These settings maximize signal and image quality without producing excessive amounts of images/data but could be adapted to different machine specifications. Other high content screening systems from PerkinElmer (e.g., Opera Phenix) are compatible with the Harmony software used herein.20.Add channels for image capture, using predefined settings:
DAPI (excitation 360–400 nm; emission 410–480 nm);Alexa 488 (excitation 460–490 nm; emission 500–550 nm);Alexa 647 and HCS Cell Mask Deep Red (excitation 620–640 nm; emission 650–760 nm).
21.Perform a z-stack image acquisition in a field of a well in order to determine the right focus plane and time of exposure for each channel (i.e., to input the right “Time” and “Height” variables for each channel). Confirm these settings are correct by taking snapshots in other wells of the plate.22.Define plate layout (typically 10–20 fields distributed in several regions of the well).23.Initiate automated image capture.**PAUSE STEP** Once images are acquired, they can be analysed at any point. Store plates at 4 °C wrapped in aluminium foil, in case additional imaging is needed.

#### 3.2.5. Image Analysis. Time for Completion: 1–2 h per Plate, per Analysis

##### 3.2.5.1. Multinucleation Analysis Sequence, Time for Completion: 1 h per Plate (Figure 3)


Input image and select “Advanced” in option Flatfield correction (Figure 3A–D).Insert “Find Nuclei” building block. Pick Channel “DAPI” and select Method “C”. In the inset menu, define Common threshold at 0.40; Area > 30 µm^2^; Split factor: 27.5; Individual Threshold: 0.40; Contrast > 0.10. Output Population: “Nuclei” (Figure 3E).Insert “Calculate Morphology Properties” building block. Select Population “Nuclei”; Region: “Nucleus”; Method “Standard”, tick “Area” and Roundness” in the inset menu. Define Output Properties as “Nuclei Properties” (Figure 3F).Insert “Select Population” building block. Select Population “Nuclei”; Method “Filter by Property” and enter as conditions “Nuclei properties Area [µm^2^] > 80” and “Nuclei properties Roundness” > 0.6. Define Output Population as “Nuclei Selected” (Figure 3G).Insert “Select Population” building block. Select Population “Nuclei Selected”; Method “Common Filters” and tick “Remove Border Objects” box. Output Population: “Nuclei Final” (Figure 3H).
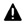
**CRITICAL STEP** These steps ensure exclusion of debris, cell clumps, and cells in the border of the field from the subsequent analysis. The exact values were optimized for hPSC-CMs and may need to be adapted to different cardiomyocyte source being investigated.Insert “Calculate Morphology Properties” building block. Select Population “Nuclei Final”; Region: “Nucleus”; Method: “Standard”. Tick “Area” and “Roundness” boxes in the inset menu. Output properties: “Nucleus Morphology”. This block does the same operation as in 3 but for the population “Nuclei Final”.Insert “Modify Population” building block. Select Population “Nuclei Final”; Region “Nucleus”; Method “Cluster by Distance”. Enter in the inset menu Distance “5 µm”; Area > “80 µm^2^”. Output population: “Total population” (Figure 3I).
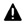
**CRITICAL STEP** These steps enable the identification of several nuclei belonging to the same cell, other than being part of the different cells, by inferring on their proximity. The values pertaining “Distance” and “Area” may need to be changed when using cardiomyocyte sources other than hPSC-CMs (i.e., displaying different morphology properties).Insert “Calculate Properties” building block. Population “Total population”; Method “By Related Population”. In the inset menu: Related Population “Nuclei final”; tick box: “Number of Nuclei Final”. Output Properties: “per Cell”. This step will effectively calculate the number of nuclei belonging to each cell.Insert “Select Population” building block. Population “Total population”; Method “Filter by Property”. Inset menu: “Number of Nuclei Final – per Cell = 1”. Output Population “Mononucleated” (Figure 3J).Insert “Select Population” building block. Population “Total population”; Method “Filter by Property”. Inset menu: “Number of Nuclei Final – per Cell = 2”. Output Population “Binucleated” (Figure 3K).


Insert “Select Population” building block. Population “Total population”; Method “Filter by Property”. Inset menu: “Number of Nuclei Final – per Cell > 2”. Output Population “Multinucleated” (Figure 3L).


11.In the default “Define Results” building block select Method ”List of Outputs” in the drop down menu, then tick the box “Number of Objects” for the options: “Population: Total population”, “Population: Mononucleated”, Population: Binucleated”, and “Population: Multinucleated”.12.In the same drop-down menu within “Define Results”, define three sections of “Method—Formula Output”. In each of these sections in the inset menu, enter “Formula: (a/b)×100; Variable A: “Mononucleated—Number of Objects”, “Binucleated—Number of Objects”, or “Multinucleated—Number of Objects”. Enter as Variable B “Total population—Number of Objects”. The Output Names for each section are “%mononucleated”, “%binucleated”, and “%multinucleated”, respectively.13.Copy the percentages pertaining to each proportion of mono-, bi-, or multinucleated cells to a GraphPad “Grouped” Table, by inserting “% mononucleated”, “% binucleated”, and “% multinucleated” in columns and different cell lines/conditions in rows.14.Create a “Stacked bars” Graph Type. Colour each bar and perform statistical analysis according to the experimental design (Figure 8C).**OPTIONAL STEP** As a quality control measure, the cardiomyocyte purity in the cell population being analysed can be determined in a separate analysis script, as follows (Figure 4):15.After repeating building blocks in steps 1–5 of Section 3.2.5.1, insert “Find Cytoplasm” building block. Channel “DRAQ5” (same excitation/emission parameters as HCS Cell Mask DeepRed); Nuclei “Nuclei Final”; Method “F”. In the inset menu, define Membrane Channel “DRAQ5”; Individual Threshold: 0.04. This step delineates the cytoplasm of cardiomyocytes by using the Cell Mask counterstain (Figure 4A–C).16.Insert “Calculate Intensity Properties” building block. Channel “Alexa 488”; Population: “Nuclei Final”; Region “Cytoplasm”; Method “Standard”, tick box “Mean” in the inset menu. Output Properties “Intensity alpha-actinin”.17.Insert “Select Population” building block. Population “Nuclei Final”; Method “Filter by Property”. In the inset menu, insert “Intensity alpha-actinin Mean” and set it > the value of the secondary only or cell-type control (Figure 4D and Figure A1). Output population “Cardiomyocytes”. Confirm if the highlighted cells display positive α-actinin signal (Figure 4E).18.Insert “Select Population” building block. Population “Nuclei Final”; Method “Filter by Property”. In the inset menu, insert “Intensity alpha-actinin Mean” and set it ≤ the value of the secondary only or cell-type control. Output population “Non-Cardiomyocytes”. Confirm if the highlighted cells display negative alpha-actinin signal (Figure 4F).19.In the default “Define Results” building block, select Method ”List of Outputs”, then tick the box “Number of Objects” for the options “Population: Nuclei Final” and “Population: Cardiomyocytes”.20.In the same drop-down menu within “Define Results”, define “Method—Formula Output”, and in the inset menu, enter “Formula: (a/b)*100; Variable A: “Cardiomyocytes—Number of Objects”; Variable B “Nuclei Final—Number of Objects”. Output Name “% Cardiomyocyte Purity”.21.Import purity values to GraphPad (e.g., “Column” table, “Scatter plot” graph) and perform statistical analysis according to the experimental design (Figure 8D). Cardiomyocyte purities should typically be >90% using directed differentiation protocols from hPSCs [35].


###### 3.2.5.2. Hypertrophic Marker BNP Analysis Sequence, Time for Completion: 2 h per Plate (Figure 5)


Input image and select “Advanced” in option Flatfield correction (Figure 5A–C).Repeat steps 2–5 from Section 3.2.5.1 to ensure exclusion of debris, cell clumps, and cells in the border of the field. Output Population: “Nuclei Final” (Figure 5D,E)Insert “Find Cytoplasm” building block. Pick Channel “DRAQ5” (same excitation/emission parameters as Alexa Fluor 647) and select Method “F”. In the inset menu, define Membrane Channel “DRAQ5”; Individual Threshold: 0.04 (Figure 5F). The sarcomeric signal is ubiquitous in cardiomyocytes so can be used to define cytoplasm, similar to the Cell Mask.Insert “Calculate Intensity Properties” building block. Select Population “Nuclei Final”; Region: “Cytoplasm”; Method “Standard”, tick “Mean” and “Standard Deviation” in the inset menu. Define Output Properties as “Intensity cardiac Troponin T”.Insert “Select Population” building block. Population “Nuclei Final”; Method “Filter by Property”. In the inset menu, insert “Intensity cardiac Troponin T” and set it > the value of the secondary only or cell-type control (e.g., fibroblasts). Output population “Cardiomyocytes”. Confirm if the highlighted cells display positive cardiac Troponin T signal. This is important so the subsequent analysis is performed only in cardiomyocytes. It can also be used for calculating purity (Figure A1 and Figure 5G).Insert “Select Region” building block. Select Population “Nuclei Final”; Region “Nucleus”; Method “Resize Region [%]”. In the inset menu, enter “-75%” in the section Outer Border. Leave “Inner Border” section empty. Output Population: “Cytoplasmic ring”. This step defines a perinuclear ring where BNP expression is maximal (Figure 5H).Insert “Calculate Intensity Properties” building block. Select Channel “Alexa 488”; Population “Cardiomyocytes”; Region: “Cytoplasmic Ring”; Method “Standard”, tick “Mean” and “Standard Deviation” in the inset menu. Define Output Properties as “Intensity BNP perinuclear”.Insert “Select Population” building block. Population “Cardiomyocytes”; Method “Filter by Property”. In the inset menu, insert “Intensity BNP perinuclear” and set it > the value of the bosentan-treated cardiomyocytes (Figure A2). Output population “BNP positive cardiomyocytes”. Confirm if the highlighted cells display positive BNP signal. This signal should always be higher than that of secondary-only or cell-type controls (Figure 5I).Insert “Select Population” building block. Population “Cardiomyocytes”; Method “Filter by Property”. In the inset menu, insert “Intensity BNP perinuclear” and set it ≤ the value of the bosentan-treated cardiomyocytes (same value as in 8). Output population “BNP negative cardiomyocytes”. Confirm if the highlighted cells do not display positive BNP signal.**OPTIONAL STEP** Further division of positive BNP signal into medium vs. high intensity can be accurately done provided ET1 and BOS treatment of cardiomyocytes has been performed, as follows.
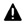
**CRITICAL STEP** Prior culture of cardiomyocytes in serum-containing medium may mask hypertrophic phenotypes [17], so exposure to serum should be minimized or fully eliminated.Insert “Select Population” building block. Population “Cardiomyocytes”; Method “Filter by Property”. In the inset menu, insert “Intensity BNP perinuclear” and set it > the value of the ET1-treated cardiomyocytes. Output population “BNP high cardiomyocytes”. Confirm if the highlighted cells display positive and high BNP signal (Figure 5J).Insert “Select Population” building block. Population “Cardiomyocytes”; Method “Filter by Property”. In the inset menu, insert “Intensity BNP perinuclear” and set it ≤ the value of the endothelin-treated cardiomyocytes. Add an additional row and set “Intensity BNP perinuclear” to ≥ the value of bosentan-treated cardiomyocytes. Output population “BNP medium cardiomyocytes”. Confirm if the highlighted cells display positive BNP signal (Figure 5K).
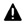
**CRITICAL STEP** Defining the threshold for positive BNP signal should be done carefully and according to the controls used. The control where cardiomyocytes were treated with ET1 should typically display >90% BNP-positive cardiomyocytes, so the intensity of BNP signal should be higher in those wells (this should be selected as the BNP-medium/high signal threshold). In contrast, >90% BOS-treated cardiomyocytes should display negative BNP signal, so the signal threshold should distinguish between negative and medium/positive BNP signal (Figure A2). The threshold for medium/high signal intensity is typically 4 times higher than that of negative/medium signal intensity (Figure 5L).In the default “Define Results” building block, select Method ”List of Outputs” in the drop down menu, then tick the box “Number of Objects” for the options: “Population: Cardiomyocytes”, “Population: Nuclei Final”, “Population: BNP positive Cardiomyocytes”, “Population: BNP negative Cardiomyocytes”, “Population: BNP medium Cardiomyocytes”, and “Population: BNP high Cardiomyocytes”.In the same drop-down menu within “Define Results”, define five sections of “Method—Formula Output”. In each of these sections in the inset menu, enter “Formula: (a/b)*100.In the first Menu define Variable A: “Cardiomyocytes—Number of Objects”, and Variable B: “Nuclei Final—Number of Objects”. Output Name: “% Cardiomyocyte Purity”.In the second menu: Variable A: “BNP positive cardiomyocytes—Number of Objects”, and Variable B “Cardiomyocytes—Number of Objects”. Output Name: “%BNP positive cardiomyocytes”.In the third menu: Variable A: “BNP high cardiomyocytes—Number of Objects”, and Variable B “Cardiomyocytes—Number of Objects”. Output Name: “%BNP high cardiomyocytes”.In the fourth menu: Variable A: “BNP medium cardiomyocytes—Number of Objects”, and Variable B “Cardiomyocytes—Number of Objects”. Output Name: “%BNP medium cardiomyocytes”.In the fifth menu: Variable A: “BNP negative cardiomyocytes—Number of Objects”, and Variable B “Cardiomyocytes—Number of Objects”. Output Name: “%BNP negative cardiomyocytes”.Check that the percentages pertaining to each Output add up. If not, confirm that the intensity thresholds between building blocks are consistent and that the comparison operators are complementary.
%BNP positive cardiomyocytes+%BNP negative cardiomyocytes = 100%%BNP positive cardiomyocytes= %BNP high cardiomyocytes+%BNP medium cardiomyocytes% BNP high cardiomyocytes+%BNP medium cardiomyocytes+%BNP negative cardiomyocytes = 100%
Import purity and %BNP positive values to GraphPad and perform statistical analysis according to the experimental design.Copy the percentages pertaining to each proportion of BNP high, BNP medium, or BNP negative cardiomyocytes cells to a GraphPad “Grouped” Table, by inserting “% high”, “% medium”, and “% negative” in columns and different cell lines/conditions in rows.Create a “Stacked bars” Graph Type. Colour each bar and perform statistical analysis according to the experimental design (Figure 8E).


###### 3.2.5.3. Sarcomeric Disarray Analysis Sequence, Time for Completion: 2.5 h per Plate (Figure 6)


Input image and select “Advanced” in option Flatfield correction (Figure 6A–C).Insert “Filter Image” building block. Pick Channel “Alexa 488”; Method “Sliding Parabola”. In the inset menu, enter value “10” in Curvature. Output Image: “Alpha-actinin sharp” (Figure 6D).**OPTIONAL STEP** The rest of this script is performed using the sharpened alpha-actinin-Alexa 488 signal, but the same principle is applicable to cardiac troponin T, as both indicate sarcomeric proteins. This can be used to confirm if the results obtained in one channel are corroborated by the other channel/sarcomeric protein.Repeat steps 2–5 from Section 3.2.5.1 to ensure exclusion of debris, cell clumps, and cells in the border of the field. Output Population: “Nuclei Final” (Figure 6E,F)Insert “Find Cytoplasm” building block. Pick Channel “DRAQ5” (same excitation/emission parameters as Alexa Fluor 647) and select Method “F”. In the inset menu, define Membrane Channel “DRAQ5”; Individual Threshold: 0.04 (Figure 6G).Insert “Calculate Intensity Properties” building block. Select Population “Nuclei Final”; Region: “Cytoplasm”; Method “Standard”, tick “Mean” and “Standard Deviation” in the inset menu. Define Output Properties as “Intensity alpha-actinin”.Insert “Select Population” building block. Population “Nuclei Final”; Method “Filter by Property”. In the inset menu, insert “Intensity alpha-actinin” and set it > the value of the secondary only or cell-type control. Output population “Cardiomyocytes”. Confirm if the highlighted cells display positive alpha-actinin signal. This is important so the subsequent analysis is performed only in cardiomyocytes. It can also be used for calculating purity (Figure A1 and Figure 6H).Insert “Calculate Morphology Properties” building block. Select Population “Nuclei Final”; Region “Cytoplasm”; Method “STAR”. In the inset menu, tick Channel “A488 sharp” and tick all the boxes in that section. In the subsection Profile Inner Region, select “Cytoplasm” and tick the box; set Profile Width to 4 px. Output Properties: “Sarcomere Morphology” (Figure 6I).Insert “Calculate Texture Properties” building block. Select Channel “A488 sharp”; Population “Nuclei Final”; Region “Cytoplasm”; Method “SER features”. In the inset menu, set Scale to “0.5 px”, Normalization by “Kernel”. Tick all the SER boxes below. Output Properties: “Sarcomere Texture SER”.Insert “Calculate Texture Properties” building block. Select Channel “A488 sharp”; Population “Nuclei Final”; Region “Cell”; Method “Haralick features”. In the inset menu, set Distance to “1 px”, and tick all the Haralick boxes below. Output Properties: “Sarcomere Texture Haralick”.Insert “Calculate Texture Properties” building block. Select Channel “A488 sharp”; Population “Nuclei Final”; Region “Cell”; Method “Gabor features”. In the inset menu, set Scale to “4 px”, Wavelength 2, Number of angles: 6, and Normalization by “Kernel”. Tick all the Gabor boxes below. Output Properties: “Sarcomere Texture Gabor”.
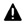
**CRITICAL STEP** Steps 7–10 are performed in order for the Harmony software to detect differences in morphology and texture between cardiomyocytes. Texture information is extracted from each image by using different filtering strategies. The parameters for each approach were optimized for hPSC-CMs and should be adapted for other cardiomyocyte sources.
Gabor: Filtering by frequency and orientation of pixels in an image;Haralick: Not based on pixel-wise filtering; each feature is a statistical characteristic of the whole image/region of interest;SER: Gaussian derivative filtering.
Insert “Select Population” building block. Population: “Cardiomyocytes”; Method “Linear Classifier”. In the inset menu, set Number of Classes to 2 and tick all the boxes below. Output Population A: “Organised”; Output Population B: “Disarrayed”.Provide a training set in the option “Training” by picking cells belonging to each class (either organized or disarrayed), in independent fields, wells, or plates. Select only cells that belong to one of these categories clearly (Figure 6J).
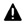
**CRITICAL STEP** Due to the heterogeneity of hPSC-CMs, a large training set of at least 100 cells for each category is desired. Phenotypic controls such as cardiomyocytes with disrupted sarcomeres (genetic knockouts or pharmacologically induced) greatly support this distinction.Evaluate the validity of the training performed by checking that the “Goodness” Parameter is >1, indicating a good separation between the 2 classes, and which texture or morphology properties more accurately define those differences (Figure A3 and Figure 6K,L).In the default “Define Results” building block, select Method ”List of Outputs” in the drop down menu, then tick the box “Number of Objects” for the options: “Population: Cardiomyocytes”, “Population: Nuclei Final”.In the same drop-down menu within “Define Results”, define two menus of “Method—Formula Output”, and in the inset menu, enter “Formula: (a/b)×100.In the first Menu define Variable A: “Cardiomyocytes—Number of Objects”; Variable B “Nuclei Final—Number of Objects”. Output Name “% Cardiomyocyte Purity”.In the second Menu, define Variable A: “Disarrayed—Number of Objects” and Variable B: “Cardiomyocytes—Number of Objects. Output Name “% Cardiomyocyte Disarray”.In the third Menu, define Variable A: “Organized—Number of Objects” and Variable B: “Cardiomyocytes—Number of Objects. Output Name “% Cardiomyocyte Organized”. Confirm that % organized + % disarrayed = 100%Import percentages to a GraphPad “Column” Table, create a “Box-plot” graph, and perform statistical analysis according to experimental design (Figure 8F).


### 3.3. Mitochondrial Respiration and Content. Time for Completion: 5 Days

#### 3.3.1. Evaluation of Mitochondrial Respiration by Seahorse™ Assay. Time for Completion: 5 Days

##### 3.3.1.1. Replating Cardiomyocytes. Time for Completion: 2 h

Coat a Seahorse XF96 cell culture microplate (cell plate) with Vitronectin following steps 1–4 from Section 3.2.1.Plate cardiomyocytes at approximately 5000 cells/mm^2^ (50,000 cells per well).
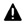
**CRITICAL STEP** This cell density is aimed at covering the well completely without forming clumps or cell aggregates (crucial for an accurate Seahorse assay and subsequent normalization). It may need to be adjusted to different cardiomyocyte sources (e.g., tissue-derived or resulting from various differentiation protocols that produce cells with varying sizes).
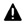
**CRITICAL STEP** Prior culture of cardiomyocytes in serum-containing medium may mask hypertrophic phenotypes [17], so exposure to serum should be minimized or fully eliminated.**OPTIONAL STEP** Allow the cells to settle for 0.5 h at room temperature before placing them in the incubator, to minimize edge effects due to the formation of convection currents because of temperature differences between the medium and the incubator.

##### 3.3.1.2. Preparation for Seahorse Assay. Time for Completion: 4 Days

3.Replace cardiomyocyte medium (100 μL/well) every 2 days for 4 days.4.On the day prior to the assay, add 200 μL of XF calibrant per well to a sensor cartridge plate (utility plate) and incubate overnight in the non-CO_2_ oven at 37 °C, to hydrate the sensors.

##### 3.3.1.3. Performing Seahorse Mitostress Test. Time for Completion: 4 h


5.Prepare XF assay medium following the kit’s instructions: 97 mL XF DMEM medium + 1 mL XF 1.0 M Glucose + 1 mL XF 100 mM Pyruvate + 1 mL XF 200 mM Glutamine. Adjust pH to 7.4. This is enough for 1 plate.6.Warm medium in water bath (37 °C). Aspirate cardiomyocyte medium and wash twice with 200 μL of XF assay medium per well. Add 175 μL medium/well. Incubate cells in non-CO_2_ oven at 37 °C for 1 h.7.Prepare injection compounds in XF assay (3 mL per compound):
Oligomycin: 12 μM to achieve a final concentration (in well) of 1.5 μM (8X dilution);FCCP: 4.5 μM to achieve a final concentration (in well) of 0.5 μM (9X dilution);Rotenone: 10 μM to achieve a final concentration (in well) of 1 μM (10X dilution).


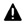
**CRITICAL STEP** These concentrations have been optimized for hPSC-CMs and may need to be adjusted to different cardiomyocyte sources8.Load the injection compounds (25 μL/port) into the cartridge plate ports using the ultrafine 200 μL tips and the corresponding port guides (port A—oligomycin; port B—FCCP; port C—rotenone; port D—XF assay medium).9.Initiate Seahorse analyser and Wave software and set up experimental protocol based on 3 basal rate measurements prior to the first injection, followed by 3 measurements after each injection. Mix-Wait-Measure times per measurement are 3 min–0 min–3 min (Figure 7A).10.Insert cartridge into the XF analyser for calibration. When prompted by the software, replace utility plate with cell plate, and run the assay to determine oxygen consumption rate (OCR). Save analysis file upon completion.


##### 3.3.1.4. Normalization of OCR Values. Time for Completion: 2 h

11.Aspirate XF medium, wash with 50 μL/well PBS and incubate with 50 μL/well Hoechst 33,342 (1:400 in PBS) for 30 min at 37 °C. Wash with PBS once and add 200 μL/well PBS.**OPTIONAL STEP** Alternatively, fix the cardiomyocytes by aspirating XF medium, washing with 50 μL/well PBS and adding 100 μL of 4% PFA per well. Incubate for 15 min at RT, wash with PBS, and add 200 μL PBS/well.
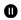
**PAUSE STEP** Fixed cells can be stored at 4 °C for up to 2 months with plates wrapped in parafilm, prior to staining and imaging.12.Image plates in Cellavista plate reader by optimizing Hoechst signal intensity and focus in a single field followed by scanning the whole plate across the entire well (Figure 7B).13.Quantify number of cell nuclei in the “Evaluation” mode. Export data to an Excel file.
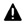
**CRITICAL STEP** Other normalization techniques based on protein content should be avoided as a hypertrophic phenotype characteristic of HCM is typically associated with increased protein/DNA ratio [38].**OPTIONAL STEP** Using Cellavista ensures high-throughput quantification of cell nuclei but other instruments (automated or not) can be used for the same purpose.

##### 3.3.1.5. Data analysis. Time for Completion: 2 h

14.Paste the cell count number of each well in the normalization tab of the Analysis file created in the Wave software upon the conclusion the Seahorse assay.15.Export data as Excel and transform into OCR values per 30,000 cells (to enable comparisons between experimental replicates and with the literature) (Figure 7C).**OPTIONAL STEP** Alternatively, input OCR and cell number values into the Excel Macro provided by Agilent for the analysis (https://www.agilent.com/en/products/cell-analysis/cell-analysis-software/data-analysis/seahorse-xf-cell-mito-stress-test-report-generators).16.Calculate parameters of mitochondrial respiration as follows (Figure 7C):
Basal respiration = average OCR [0−15min]−average OCR [80−95min]Maximal respiration = average OCR [40−55min]−average OCR [average OCR 60−75min]ATP production = average OCR [0−15min]−average OCR [20−35min]
**OPTIONAL STEP** Additional parameters can also be reported although these are not as informative for phenotyping HCM:
Spare capacity = average OCR [40−55min]−basal respirationNon−mitochondrial respiration = average OCR [60−75min]Proton leak = OCR [20−35min]−average OCR [60−75min]
17.Import data into GraphPad as an “XY Table”. Plot and amend the colour of the lines/bar graphs according to the experimental design and indicate in the graph when compounds were added (Figure 8G–J).18.Perform statistical analysis (depending on the experimental design).

#### 3.3.2. Evaluation of Mitochondrial Content by qPCR. Time for Completion: 1 Day

##### 3.3.2.1. DNA Extraction from Cardiomyocytes. Time for Completion: 1 h

Pellet cardiomyocyte single-cell suspension by centrifugation at 100 xg for 15 min in a 1.7 mL tube.Aspirate supernatant. Wash with 1 mL PBS and pellet again.Extract DNA using Qiagen Blood and Tissue kit, following manufacturers’ instructions.Determine DNA concentration and purity using NanoDrop or equivalent.

##### 3.3.2.2. Quantitative Real-Time PCR for Mitochondrial/Nuclear DNA Ratio. Time for Completion: 4 h

5.Dilute DNA samples to 25 ng/μL in molecular-grade water.6.Prepare qPCR mastermixes as follows (per reaction):
Mastermix: 5 μL;Probe (ACTB, MT-ND1 or MT-ND2): 0.5 μL;Molecular grade water: 3.5 μL.
7.Reverse-pipette 9 μL of each reaction mastermix into the MicroAmp™ Fast Optical 96-Well Reaction Plate.
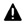
**CRITICAL STEP** Reverse-pipetting is to ensure high accuracy of pipetting technique, due to the very high sensitivity of the qPCR technique.8.Add 1 μL of DNA template to each respective well of the 96-well plate.9.Stick the MicroAmp™ Optical Adhesive Film to the 96-well plate.10.Initiate the 7500 Real-Time PCR equipment and its respective software. Define experimental setup as: Instrument “7500 Fast (96 Wells); Type of Experiment “Quantitative – Comparative C_T_ (ΔΔC_T_); Reagents “TaqMan^®^ Reagents”; Ramp Speed “Standard (~2 h to complete a run).11.Enter plate setup according to experimental design. Set Run Method as:
Holding stage: 50.0 °C for 20 s, followed by 95.0 °C for 10 min;Cycling stages: 95.0 °C for 15 s, followed by 60.0 °C for 1 min—40 cycles.
12.Run qPCR reaction. Save data upon completion.13.Analyse data using the ΔΔCT method, where the average of healthy cardiomyocytes is used for calculating relative quantification of mtDNA content (i.e., used as the control sample) [39]:
$$\Delta C_{T}mitoDNAgene,\;sampleX\;=\;C_{T}mitoDNAgene,\;sampleX\;-\;C_{T}nuclearDNAgene,sampleX\;\Delta \Delta C_{T}mitoDNAgene,sampleX=\;\Delta C_{T}mitoDNAgene,\;sampleX\;-\;\Delta C_{T}mitoDNAgene,control\;sample\;\mathrm{Expression}\;\mathrm{fold}\;\mathrm{change}\;\mathrm{mitoDNAgene},\;\mathrm{sampleX}=2^{\Delta \Delta C_{T}\mathrm{mitoDNAgene},\mathrm{sampleX}}$$
14.Import data into GraphPad. Perform statistical analysis according to the experimental design (Figure 8K).

## 4. Expected Results

HCM is a complex and intractable disease that requires further characterization to facilitate therapeutic intervention [2]. Several cellular and molecular mechanisms of disease have been described [38], but they have shown to be mutation-specific and/or restricted to particular patient backgrounds [10]. Thus, uncovering novel molecular mechanisms of HCM progression in cellular models where the genetic causation has been elucidated is a promising strategy to address the lack of effective treatment in order to restore cardiac function [4,40].

So as to address the complex in vitro characterization of HCM, we developed a set of phenotypic assays that can be used to quantify disease hallmarks. Our toolkit consists of three straightforward protocols with varying degrees of logistic requirement (e.g., equipment and cost) that are not only accessible to any research groups investigating HCM, but also applicable to other fields evaluating identical changes in cellular responses. The data generated from this characterization encompass analysis of (i) hypertrophy, (ii) HCM molecular features, and (iii) mitochondrial respiration and content (Figure 8).

While cardiomyocytes cultured in 2D conditions do not fully replicate pathophysiological disease features [29,41], the information acquired from such a multi-parametric evaluation produces a significant level of understanding of HCM characteristics with directly quantifiable variables. Remarkably, these can be harnessed in high-throughput drug screening approaches aimed at identifying new compounds with therapeutic potential, which are integrated in early stages of drug discovery pipelines where fast and inexpensive assays are paramount [6]. In this regard, this toolkit can be used to detect attenuation or rescue of HCM phenotypes upon pharmacological of genetic intervention. This is useful to exclude drug candidates early on, before more complex, lower-throughput, and often time-consuming and labour-intensive assays are required [5]. This greatly expedites subsequent functional assaying of cardiomyocytes (e.g., investigating calcium handling and contractile force), typically performed in 3D organoids or engineered heart tissues [26,27,28,42,43,44].

Overall, it is expected that disseminating this toolkit will greatly support disease modelling research fields, accelerating the discovery of new mechanisms of disease and ultimately paving the way for more efficient therapeutics.

## Figures and Tables

**Figure 1 mps-02-00083-f001:**
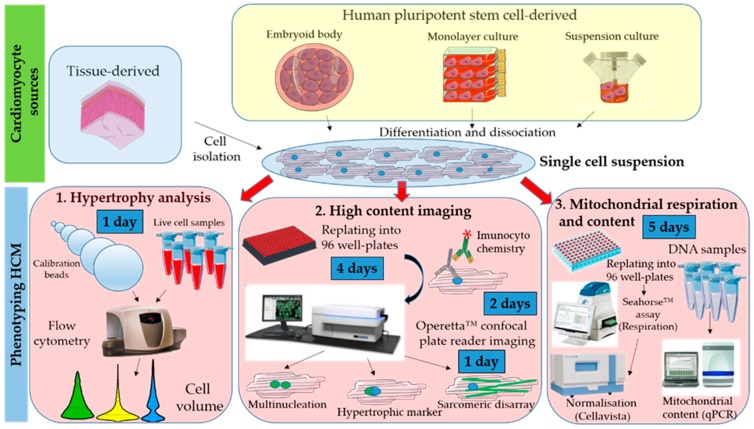
Workflow showing toolkit for phenotyping hypertrophic cardiomyopathy (HCM) in cellular models. Cardiomyocytes can be obtained from different sources (biopsies or cell lines) and require dissociation into a single cell suspension prior to phenotyping. Evaluation of in vitro hallmarks of HCM is done via: (1) Flow volumetry to investigate hypertrophy; (2) high-content imaging to assess multinucleation, brain natriuretic peptide (BNP) hypertrophic marker expression, and sarcomeric disarray; (3) Seahorse assay and qPCR to assess mitochondrial respiration and content, respectively.

**Figure 2 mps-02-00083-f002:**
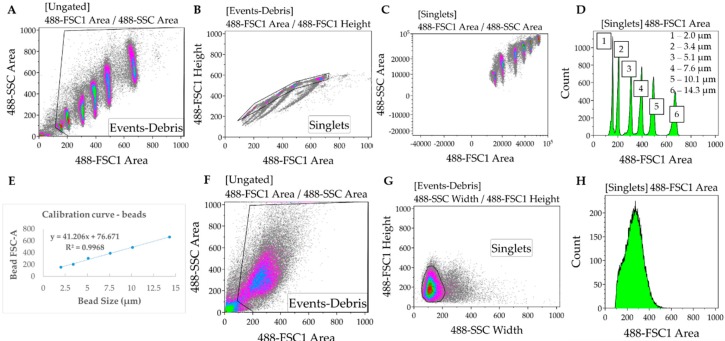
Protocol for evaluating hypertrophy by flow cytometry. (**A**–**C**) Forward (FSC) and side (SSC) scatter values of calibration beads of known sizes are measured, and debris and clumps are excluded by gating. (**D**,**E**) Each bead has a distinctive FSC value that is used to plot a calibration curve (FSC vs. size). (**F**–**H**) Cardiomyocyte samples are analysed and gated similarly to calibration beads to measure FSC values, which are used to determine cell volume.

**Figure 3 mps-02-00083-f003:**
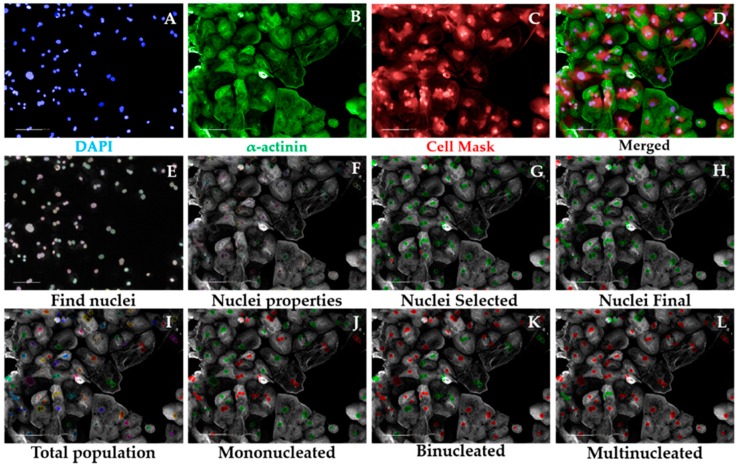
Multinucleation Analysis Sequence by high-content imaging. Representative micrographs of image analysis of human pluripotent stem cell-derived cardiomyocytes (hPSC-CMs). After inputting the image (**A–D**), the software excludes debris and cell clumps (**E–H**) and identifies different degrees of multinucleation based on the distance between nuclei in cardiomyocytes (**I–L**). Scale bar = 100 μm.

**Figure 4 mps-02-00083-f004:**
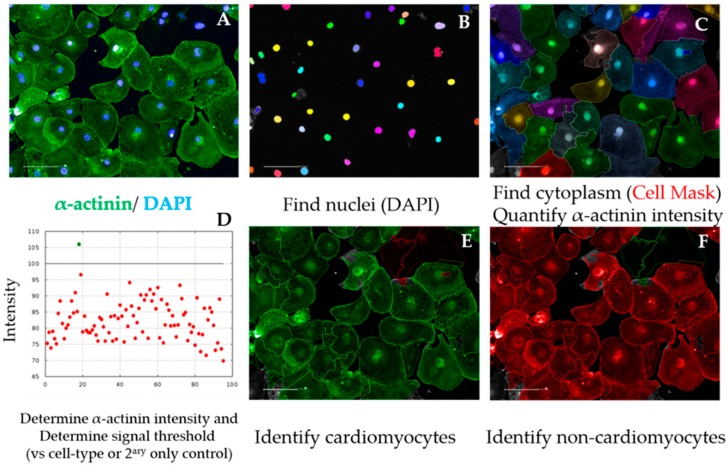
Cardiomyocyte purity analysis sequence. Representative micrographs of image analysis of hPSC-CMs. After inputting the image (**A**), the software excludes debris and cell clumps (**B**) and uses the cell mask staining to define cell cytoplasm (**C**). The α-actinin signal intensity is quantified in the cytoplasm and a positive signal threshold is determined by comparing with cell-type control (e.g., fibroblasts) (**D**), enabling the distinction of cardiomyocytes (**E**) from non-cardiomyocytes (**F**), required to calculate purity. Scale bar = 100 μm.

**Figure 5 mps-02-00083-f005:**
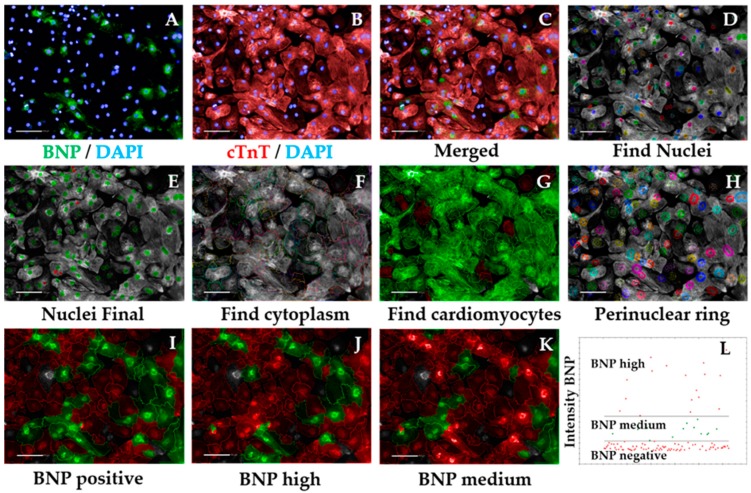
Hypertrophic marker BNP analysis sequence. Representative micrographs of image analysis of hPSC-CMs. After inputting the image (**A**–**C**), the software excludes debris and cell clumps (**D**,**E**) and uses the cardiac troponin T (cTnT) staining to define cell cytoplasm (**F**). The cTnT signal intensity is quantified and a positive signal threshold is determined by comparing with cell-type control enabling exclusion of non-cardiomyocytes from the analysis (**G**). A perinuclear ring is defined in cardiomyocytes (**H**) and BNP intensity is quantified in this region. Comparison of BNP signal intensity to that of ET1- or BOS-treated cardiomyocytes enables distinction between high vs. medium BNP-expressing cells (**I**–**L**). BNP—brain natriuretic peptide; cTnT—cardiac troponin T; scale bar = 100 μm.

**Figure 6 mps-02-00083-f006:**
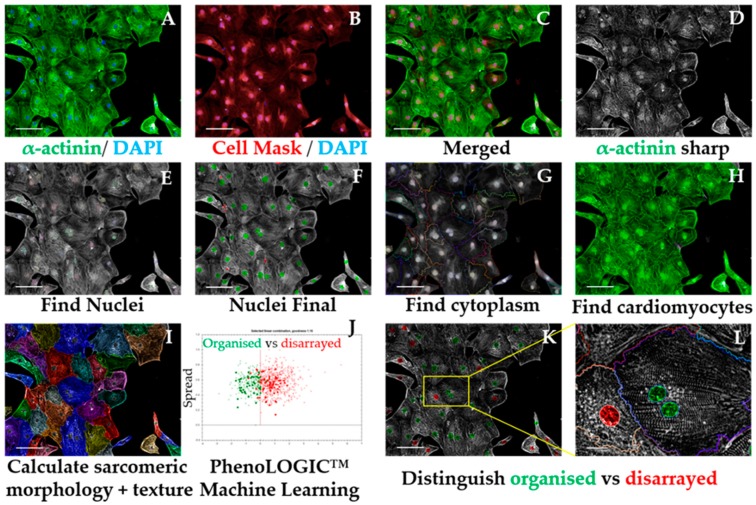
Sarcomeric disarray analysis sequence. Representative micrographs of image analysis of hPSC-CMs. After inputting the image (**A–C**), the software uses mathematical transformation algorithms to sharpen sarcomeric signal (**D**), followed by exclusion of debris and cell clumps (**E,F**). The cell mask channel is used to define cell cytoplasm (**G**) and the α-actinin signal intensity is quantified and compared against that of a cell-type control to exclude non-cardiomyocytes from the analysis (**H**). Morphologic and texture-based information is extracted from the sharpened α-actinin signal displayed by cardiomyocytes (**I**), followed by a machine-learning procedure where the user manually selects cardiomyocytes displaying organised vs. disarrayed sarcomeres (**J**). The software then applies this algorithm to the rest of the images being analysed to distinguish organised vs. disarrayed cardiomyocytes (**K,L**). Scale bar = 100 μm, except for L where it is 20 μm.

**Figure 7 mps-02-00083-f007:**
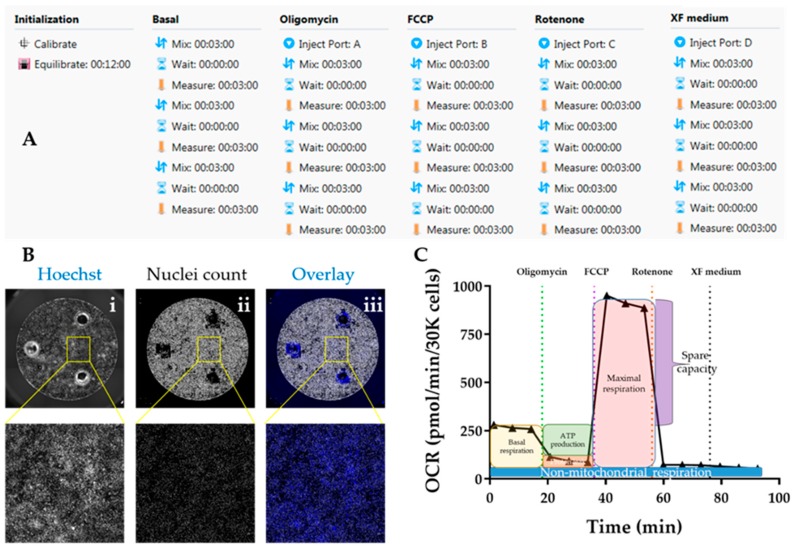
Evaluation of mitochondrial respiration using Seahorse platform. (**A**) Protocol summary for mitostress test using the Seahorse assay™. (**B**) Representative micrographs detailing normalisation by automated quantification of cell nuclei in Cellavista™ plate imager ((i) Hoechst staining of nuclei; (ii) quantification of individual nuclei; (iii) overlay). (**C**) Sample graph for oxygen consumption rate (OCR) over time following the Seahorse mitostress kit, highlighting the different mitochondrial respiration parameters.

**Figure 8 mps-02-00083-f008:**
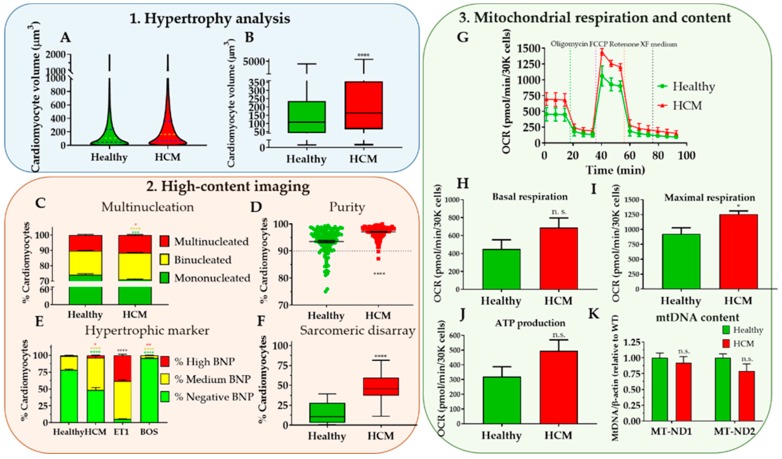
Example of application of phenotyping toolkit to HCM cellular model. (**A**,**B**) Hypertrophy analysis by flow volumetry indicating increased median volume in HCM cardiomyocytes relative to healthy counterparts. (**C**–**F**) High-content imaging evaluation of HCM cardiomyocytes showing increased proportion of multinucleation, expression of hypertrophic marker BNP, and sarcomeric disarray relative to healthy controls. (**G**–**J**) Assessment of mitochondrial respiration in Seahorse™ assay revealing higher energy production in HCM cardiomyocytes in comparison with healthy ones, showing (**K**) similar mitochondrial DNA content (determined by ratiometric mito/nuclear DNA qPCR analysis). Data represent mean +/- SEM of five independent experiments comparing R453C-β-myosin heavy-chain hPSC-CMs (HCM) to isogenic wild-type control (healthy), with 100,000–250,000 cells analysed per sample. Statistical analysis was performed by unpaired parametric Student’s t test between HCM and Healthy conditions, with the exception of Panel E, where unpaired one-way ANOVA test was used with Dunnett’s post hoc test for correction of multiple comparisons relative to Healthy condition (n. s., non-significant; **P* < 0.05; ***P* < 0.01; ****P* < 0.005; *****P* < 0.0001, colour-coded by inter-compared category—black asterisks apply to all categories).

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
