# Peer review of "High-Throughput Phenotyping Toolkit for Characterizing Cellular Models of Hypertrophic Cardiomyopathy In Vitro"

_mps, 2019, doi:10.3390/mps2040083_

Round 1
Reviewer 1 Report
The work by Mosqueira et al., describes three different protocols to phenotypically assess cardiomyocytes derived from hiPSCs. The topic is of interest in the scientific community studying cellular mechanisms of cardiac diseases. and this work provides details regarding a few possible approaches. I think though, they should expand the importance of other techniques to study for instance contractility, calcium and voltage transients, expression of the genetic mutants leading to the disease.These measurements can be done in 2D too.
Hoechst is misspelled in figure 7 and in its legend
Figures seem to be of low quality in the pdf file I received.
Author Response
1) The work by Mosqueira et al., describes three different protocols to phenotypically assess cardiomyocytes derived from hiPSCs. The topic is of interest in the scientific community studying cellular mechanisms of cardiac diseases and this work provides details regarding a few possible approaches. I think though, they should expand the importance of other techniques to study for instance contractility, calcium and voltage transients, expression of the genetic mutants leading to the disease. These measurements can be done in 2D too.
We agree with the Reviewer that this is an important point, and have addressed it by including the following paragraph in the Introduction section:
The information gained by these tools can be harnessed to narrow down the number of conditions (i.e., genetic mutants and/or drugs) meriting further investigation, based on the analysis of cardiomyocyte contractility, calcium handling and voltage transients. The established technologies to evaluate these functional parameters (e.g., patch clamp electrophysiology, multi-electrode arrays, optical imaging and traction force microscopy) typically rely on low- to medium-throughput assays using single cells, 3D organoid and/or engineered heart tissues [5, 26]. However, recent developments have enabled measurement of contractility across all these configurations, based on quantification of pixel displacement in high-speed movies using a publicly available software [27]. Remarkably, it is now even possible to measure action potentials, cytosolic calcium flux and contractility simultaneously [28]. These tools have achieved 44-78% drug predictivity scores in hPSC-CMs [29]. Moreover, despite the existence of challenges such as incomplete hPSC-CM maturity and lack of multicellular complexity, these approaches have been used to investigate the effects of genetic mutations in HCM progression, towards clarifying genotype-phenotype relationships [8].
2) Hoechst is misspelled in figure 7 and in its legend.
This typo has now been corrected in Figure 7 and its legend accordingly (Hoescht now reads Hoechst).
3) Figures seem to be of low quality in the pdf file I received.
All the images were provided as high-quality PNG files, both embedded in the word file and separately. We presume the low quality should be due to the PDF rendering process so we would kindly ask the Reviewer to download the figures separately as to confirm their high resolution.
Reviewer 2 Report
Summary:
In this manuscript the authors detail a set of unbiased approaches designed to characterize hypertrophic markers in cardiac myocytes derived from biopsies or iPS cells. Overall these procedures are well-described and seem valuable when used to accompany more specific investigations.
Comments:
The investigators spend little time on optimizing flow cytometry for cardiac myocytes derived from tissue biopsies. This is a tricky experiment that has been the focus of entire articles on its own. In particular, more attention should be given to procedures to break up cell clumps, whether filtering should be used, and the size of the flow cell/nozzle that is required for reproducible results. Most of the procedures are described as having been optimized for cardiac myocytes derived from iPS cell sources. If these methods are intended to be used in the characterization of both iPS cell-derived cardiac myocytes and tissue-derived cardiac myocytes, for each procedure some specific comments should be given about potential issues that may come up when using tissue-derived cardiac myocytes (see point #1, above) and how they might be solved. The example figures throughout the manuscript were quite helpful, as was Figure 8, demonstrating the expected data. However, in order to increase the utility of these figures, the investigators should treat this data as if they were reporting an experiment. That means describing the source of the cells, number of cells/isolations used in each assay, and for Figure 8, a statistical analysis of the results, etc.Author Response
The investigators spend little time on optimizing flow cytometry for cardiac myocytes derived from tissue biopsies. This is a tricky experiment that has been the focus of entire articles on its own. In particular, more attention should be given to procedures to break up cell clumps, whether filtering should be used, and the size of the flow cell/nozzle that is required for reproducible results. Most of the procedures are described as having been optimized for cardiac myocytes derived from iPS cell sources. If these methods are intended to be used in the characterization of both iPS cell-derived cardiac myocytes and tissue-derived cardiac myocytes, for each procedure some specific comments should be given about potential issues that may come up when using tissue-derived cardiac myocytes (see point #1, above) and how they might be solved.
We acknowledge the Reviewer’s concern and have provided additional information on isolating and analysing cardiomyocytes derived from tissue biopsies. The Experimental Design section has been expanded to address this point:
Efficient isolation of cardiomyocytes from heart tissue biopsies has been the subject of numerous reports and typically consists of enzymatic bulk digestion [30], Langendorff method [31] or mechanical disruption procedures [32]. A recent protocol optimized this process in five main steps: i) myocardium dissection into 200 μm-thick slices; ii) slice perfusion with a Ca2+-free solution; iii) enzymatic digestion using collagenase II and protease XXIV; iv) filtration of cardiac tissue extract with a 100 μm mesh (to break cell clumps and minimize cell sampling biases [33]); and v) in vitro culture in 5% fetal bovine serum-containing medium [34]. This method resulted in up to 65% viable cardiomyocyte isolation yield and enabled phenotypic studies of electrophysiology, Ca2+ imaging and Seahorse™ analysis. Alternatively, cardiomyocytes can be efficiently sourced in high numbers through cardiac differentiation of hPSCs [35] followed by their dissociation into single cells using a collagenase II digestion protocol [36].
Moreover, additional considerations addressing specific issues associated with the use of tissue-derived cardiac myocytes have been included:
Section 3.1.1.: CRITICAL STEP Use of 100 μm flow cytometer nozzle size is highly recommended to minimize sampling biases [33].
Sections 3.1.1.; 3.2.5.2 and 3.3.1.1.: CRITICAL STEP Prior culture of cardiomyocytes in serum-containing medium may mask hypertrophic phenotypes [17], so exposure to serum should be minimized or fully eliminated.
The example figures throughout the manuscript were quite helpful, as was Figure 8, demonstrating the expected data. However, in order to increase the utility of these figures, the investigators should treat this data as if they were reporting an experiment. That means describing the source of the cells, number of cells/isolations used in each assay, and for Figure 8, a statistical analysis of the results, etc.
We thank the Reviewer for the constructive suggestion and have amended the manuscript accordingly, as follows.
1) We have clarified the provenance of the data by including the following in the Experimental Design section:
All the exemplary data reported in this manuscript (Figure 8) is based on the comparison between an hPSC-CM cell line bearing the R453C-β-myosin heavy chain mutation (HCM line) and its isogenic wild-type control (healthy).
2) All the experimental conditions pertaining each protocol were described in detail in their respective sections, including the number of cells for each assay, for example:
3.1.1.Resuspend freshly-dissociated cardiomyocytes in 500 μl Phosphate Buffer Saline (PBS) at 1x105-2.5x105 cells per sample and place them on ice prior to flow cytometry analysis; 3.2.1. Plate cardiomyocytes at approximately 110,000 cells/cm2 (35,000 cells per well).
3) A statistical analysis of the results has now been performed as requested, Figure 8 has been corrected, and the following description has been added to its Legend:
Data represent mean +/- SEM of 5 independent experiments comparing R453C-β-myosin heavy chain hPSC-CMs (HCM) to isogenic wild-type control (healthy), with 100,000-250,000 cells analysed per sample. Statistical analysis was performed by unpaired parametric Student’s t test between HCM and Healthy conditions, with the exception of Panel E, where unpaired one-way ANOVA test was used with Dunnett’s post hoc test for correction of multiple comparisons relative to Healthy condition (n. s. – non-significant; *P < 0.05; **P < 0.01; ***P < 0.005; ****P < 0.0001, colour-coded by inter-compared category - black asterisks apply to all categories).
Round 2
Reviewer 2 Report
My concerns have been addressed by the authors.